# Chlorovirus PBCV-1 Multidomain Protein A111/114R Has Three Glycosyltransferase Functions Involved in the Synthesis of Atypical N-Glycans

**DOI:** 10.3390/v13010087

**Published:** 2021-01-10

**Authors:** Eric Noel, Anna Notaro, Immacolata Speciale, Garry A. Duncan, Cristina De Castro, James L. Van Etten

**Affiliations:** 1Nebraska Center for Virology, University of Nebraska, Lincoln, NE 68583-0900, USA; enoel@huskers.unl.edu (E.N.); gduncan2@unl.edu (G.A.D.); 2School of Biological Sciences, University of Nebraska, Lincoln, NE 68588-0118, USA; 3Department of Chemical Sciences, University of Napoli Federico II, Via Cintia 4, 80126 Napoli, Italy; anna.notaro@unina.it (A.N.); immacolata.speciale@unina.it (I.S.); 4Information Génomique et Structurale, Centre National de la Recherche Scientifique, Aix-Marseille Université, UMR 7256, IMM FR3479, Institut de Microbiologie (FR3479), CEDEX 9, 13288 Marseille, France; 5Department of Agricultural Sciences, University of Napoli Federico II, Via Università 100, 80055 Portici, Italy; 6Department of Plant Pathology, University of Nebraska, Lincoln, NE 68583-0722, USA

**Keywords:** glycosyltransferases, multidomain protein, N-glycan, chloroviruses, PBCV-1

## Abstract

The structures of the four *N*-linked glycans from the prototype chlorovirus PBCV-1 major capsid protein do not resemble any other glycans in the three domains of life. All known chloroviruses and antigenic variants (or mutants) share a unique conserved central glycan core consisting of five sugars, except for antigenic mutant virus P1L6, which has four of the five sugars. A combination of genetic and structural analyses indicates that the protein coded by PBCV-1 gene *a111/114r*, conserved in all chloroviruses, is a glycosyltransferase with three putative domains of approximately 300 amino acids each. Here, in addition to in silico sequence analysis and protein modeling, we measured the hydrolytic activity of protein A111/114R. The results suggest that domain 1 is a galactosyltransferase, domain 2 is a xylosyltransferase and domain 3 is a fucosyltransferase. Thus, A111/114R is the protein likely responsible for the attachment of three of the five conserved residues of the core region of this complex glycan, and, if biochemically corroborated, it would be the second three-domain protein coded by PBCV-1 that is involved in glycan synthesis. Importantly, these findings provide additional support that the chloroviruses do not use the canonical host endoplasmic reticulum–Golgi glycosylation pathway to glycosylate their glycoproteins; instead, they perform glycosylation independent of cellular organelles using virus-encoded enzymes.

## 1. Introduction

Glycosylation is an important post-translational modification that confers a diversity of structures and functions in both cell and virus biology. One of the most common forms of protein modification is *N*-linked glycosylation, in which a high mannose core is linked to the amide nitrogen of asparagine in the context of the conserved motif Asn–X–Ser/Thr. This attachment occurs early in protein synthesis, followed by a complex process of trimming and remodeling of the oligosaccharide during transit through the endoplasmic reticulum (ER) and Golgi [1] producing glycoproteins with varying oligosaccharide structures. Typically, viruses use host-encoded glycosyltransferases (GT) and glycosidases located in the ER and Golgi to add and remove *N*-linked sugar residues to/from virus glycoproteins either co-translationally or shortly after translation of the protein. The viral glycoproteins are then often transported to the host plasma-membrane where progeny viruses bud though the membrane, and so the viruses only become infectious as they leave the cell [2].

However, the large dsDNA chloroviruses in the family *Phycodnaviridae* are interesting because, unlike other viruses, they encode most, if not all, of the machinery to glycosylate their major capsid proteins (MCP) independent of the cellular organelles [3,4]. Furthermore, the process occurs in the cytoplasm, and infectious viruses are formed inside the cell prior to cell lysis. The prototype chlorovirus, Paramecium bursaria chlorella virus (PBCV-1), has a 330-kb genome that is predicted to encode as many as 400 proteins, many of which are unusual for a virus [5], including at least 17 related to various aspects of carbohydrate metabolism [6].

The PBCV-1 MCP (also referred to as Vp54) is coded by gene *a430l*, and the protein has four glycosylation sites [7]. The predominant oligosaccharide is a nonasaccharide (Figure 1) with several unique features, including the following: (i) its structure does not resemble any structure previously reported in the three domains of life [8]; (ii) the four glycoforms are generated by the non-stochiometric presence of two monosaccharides, L-arabinose (Ara*f*) and D-mannose (Man); (iii) the most abundant glycoform consists of nine neutral monosaccharide residues organized in a highly-branched fashion; (iv) none of the *N*-linked glycans is attached to a typical Asn–X–(Thr/Ser) consensus site in the protein [9]; (v) the glycans are attached to the protein by a β-glucose (Glc) linkage, which is rare in nature and has only been reported in glycoproteins from a few organisms [10,11,12,13]; and (vi) the glycoform contains a dimethylated rhamnose (Rha) as the capping residue of the main chain, a hyper-branched fucose (Fuc) residue and two Rha residues with opposite configurations.

Interestingly, all the chloroviruses studied to date, including those with different host specificities, have the capsid protein N-glycosylated with other types of oligosaccharides; however, all chloroviruses share the same pentasaccharide core oligosaccharide [14,15] composed of an *N*-linked Glc, a hyperbranched Fuc, a distal and a proximal xylose (Xyl), and a galactose (Gal) (Figure 1). Additional monosaccharides decorate this core N-glycan, producing a molecular signature for each chlorovirus [16]. This oligosaccharide core was also found in all mutants (or antigenic variants) of PBCV-1 analyzed to date, except for PIL6 (Figure 1 [6]). Mutant PIL6 is a representative of the antigenic class D, characterized by a large genomic deletion spanning genes *a014r* through *a078r* [6,17]. Its N-glycan is a tetrasaccharide and is significantly truncated compared to that of wild-type PBCV-1. It contains all the units of the chloroviruses N-glycans present in the conserved core region except the distal Xyl attached to Fuc [16]. By analyzing the genomes of all chloroviruses and of the mutants sequenced to date, we noted that the *a111/114r* gene is the only annotated orthologous GT gene found outside of the region of the large deletion mutants that is present in all other chloroviruses. This finding suggests its likely involvement in the synthesis of the initial part of the unusual N-glycan shared by all of these viruses and prompted us to investigate its role in the assemblage of the conserved core oligosaccharide.

To predict the A111/114R protein structure and functions, we used a combination of genetic and structural analyses, together with hydrolytic activity assays and in silico evidence by sequence analysis and protein modeling. The combined results suggest that the A111/114R protein is a multi-domain/multi-functional GT likely involved in the attachment of three of the five monosaccharides in the conserved core region of the N-glycan (Figure 1). Specifically, this large protein of 860 amino acids is comprised of three putative domains (Appendix A; Figure 2), each with a specific role; the N-terminal domain (1–260 aa) is a galactosyltransferase (GalT), the central domain (261–559 aa) is a xylosyltransferase (XylT), and the C-terminal domain (560–860 aa) is a fucosyltransferase (FucT).

## 2. Materials and Methods

### 2.1. Protein Modeling

The prediction of the different domains of A111/114R was performed by HHpred [19] based on remote homology detection. Then, the 3D model of each domain was built by Phyre2 (Protein Homology/analogY Recognition Engine V 2.0) in a normal mode [18]. Each 3D model was based on an alignment generated by HMM–HMM matching. Final drawings and residue analysis were prepared with the molecular graphics system PyMol Version 1.2r3pre, Schrödinger, LLC.

### 2.2. Cloning and Expression

PBCV-1 *a111/114r* and domain variants were cloned from PCR-amplified viral DNA using oligonucleotide primers with restriction sites NotI–BamHI. PCR fragments of the expected size were digested and inserted into the restriction sites of the pMAL-c6T expression vector (New England Biolabs, Ipswich, MA, USA). This process produced a maltose-binding protein (MBP) tag at the N-terminus of the target protein. The resulting plasmid was transformed into *E. coli* strain One Shot TOP10 competent cells (Invitrogen) for maintenance. The *E. coli* cells containing positive cloned plasmids were selected with 100 μg/mL carbenicillin. The cloned structure of each vector was sequence verified. Plasmids were isolated with a QIAprep Spin Miniprep kit (Qiagen, Valencia, CA, USA) according to the manufacturer’s instructions and transformed into NEBExpress competent cells for expression. Viral genes were expressed by growing cells overnight at 37 °C in 10 mL of LB medium (10 g/L tryptone, 5 g/L yeast extract, 5 g/L sodium chloride) containing 100 μg/mL carbenicillin. Then, 5 mL of the over-night culture was sub-cultured into 200 mL LB medium containing 100 μg/mL carbenicillin. The batch culture was grown to an OD_600_ of 0.6 at 37 °C and then induced with 0.1 mM IPTG and incubated at 16 °C overnight. The cells were harvested by centrifugation at 3500× *g*, for 5 min at 4 °C, and resuspended in 35 mL of PBS with 2 mM phenylmethylsulfonyl fluoride (PMSF). After incubation on ice for 30 min, cells were disrupted by sonication for 3 min using a Tekmar sonic disruptor at 30% amplitude, in 5 s pulses. Samples were centrifuged at 10,000 rpm for 15 min to separate soluble and insoluble fractions.

### 2.3. Purification of Recombinant Enzymes

Amylose resin (New England Biolabs) was loaded onto a 5-mL self-packing column with a 45- to 90-μm-pore-size polyethylene filter (frit) (Life Science Products, Chestertown, MD, USA), and the resin was allowed to settle. The column was equilibrated with 5 column volumes of cold wash buffer (50 mM NaH_2_PO_4_, 150 mM NaCl, 1 mM DTT, 1 mM EDTA, pH 7.2). The soluble bacterial fraction was applied to the column and allowed to drain. The column was washed again with 5 column volumes of cold wash buffer. The recombinant proteins were eluted with the MBP moiety using elution buffer (wash buffer plus 10 mM maltose). The recombinant protein concentrations were determined by a NanoDrop spectrophotometer (NanoDrop Technologies, Wilmington, DE, USA). Eluted proteins were resolved by SDS-PAGE (7.5% acrylamide) with Coomassie brilliant blue staining.

### 2.4. UDP-Glo^TM^ and GDP-Glo^TM^ GT Assays

Detection of free uridine diphosphate (UDP) after hydrolysis of the sugar nucleotide was performed using the UDP-Glo^TM^ GT assay kit (Promega Corporation, Madison, WI, USA), which detects UDP after UDP-sugar hydrolysis or transfer by converting UDP to light (measured in Relative Luminescence Units) in a luciferase-type reaction. Detection of free guanosine diphosphate (GDP) from GDP-sugar hydrolysis was evaluated using the GDP-Glo^TM^ GT assay kit (Promega Corporation), which operates by the same principles as above. A standard curve using 0–25 μM of the respective nucleoside diphosphate (NDP) was performed, and the range of measurements was determined to be in the linear range of detection, where the luminescence detected is directly proportional to NDP concentration.

Enzymes were diluted with an optimized GT solution (0.1 M MOPS-NaOH pH 7, 10 mM Mn^2+^) and supplemented with 100 μM of the targeted nucleotide sugar(s). Each sugar–nucleotide hydrolysis reaction was incubated at 16 °C for 16 h. Following the manufacturer’s protocol, each reaction was combined in a ratio of 1:1 (25 μL: 25 μL) with the UDP-Glo^TM^ detection reagent in independent wells of a white, flat bottom 96-well assay plate and allowed to incubate at ambient temperature. After 1 h of incubation, luminescence was measured in triplicate, which is directly proportional to UDP or GDP concentration based on the standard curves. Luminescence measurements were performed with a Veritas^TM^ microplate luminometer (Turner Biosystems, Sunnyvale, CA, USA) using a 96-well microplate with standard 128 × 86-mm geometry with an integration time of 1.0 s. Luminescence was measured by a low-noise photomultiplier detector through an empty filter position in the emission filter wheel.

## 3. Results and Discussion

### 3.1. In Silico Analysis of A111/114R Domain 1

PBCV-1 encoded protein A111/114R was annotated in NCBI (https://blast.ncbi.nlm.nih.gov/) as a hypothetical protein (NP_048459) of which only one region was predicted to be a GT (262 to 382 aa). PSI-BLAST analysis [20] revealed that A111\114R was conserved among all chloroviruses with 62–94% sequence identity, reinforcing the notion that it could be involved in the assembly of the conserved oligosaccharide core. Therefore, to elucidate the function of A111\114R, we analyzed the full-protein sequence using HHpred tool [19], which predicted three putative GT domains. N-terminal homology with several known GTs (>98% probability) was identified for residues 1 to 256; hence, we assigned residues 1 to 260 aa as domain 1 (A111/114R-D1). The other two domains are discussed in the following sections.

Protein structure prediction by Phyre2 analysis [18] identified six GT crystal structures (Appendix A) to model three-dimensional structures of A111/114R-D1 based on 50–82% protein coverage (>90% confidence and 12–23% sequence identity). Models were ranked according to raw alignment score using the sequence and the secondary structure similarity, inserts, and deletions. Interestingly, the top ranked models were based on two XylTs (PDB:6BSV, 4WMA), one glucosyltransferase (GlcT) (PDB:1LL2), and three GalTs (PDB:1GA8, 5GVV, 6U4B), in agreement with our initial hypothesis, as both Xyl and Gal are part of the conserved region of the core glycan (Figure 1). The highest ranked model was the XylT XXT1 from *Arabidopsis* (PDB: 6BSV) based on 120 residues (19% identity with 96% confidence). However, the homologous region of A111/114R-D1 (9 to 168 aa) that aligned with XXT1 residues 167 to 287 had no homology with the residues in the active site of the XXT1 model, which utilized Lys-382, Asp-317, Asp-318, and Gln-319 to bind the nucleotide sugar for catalytic activity [21]. Similarly, important residues for enzyme activity in the XylT XXYLT1 (PDB: 4WMA) from *Mus musculus* [22] (His-262, Trp-265, and Gly-325) did not share homologous positions with A111/114R-D1. Based on this evidence, we deduced that A111/114R-D1 has no XylT activity, and for this reason we examined the other high top ranked models, all reporting well characterized GalTs (PDB: 1GA8, 5GVV, 6U4B), except for 1LL2 which is a GlcT involved in glycogen synthesis [23]. Although GlcT activity cannot be definitely ruled out, A111/114R-D1 lacks the equivalent 1LL2 residue, Tyr-194, involved in Glc addition.

In order to investigate possible GalT activity for A111/114R-D1, we chose, as reference protein, LgtC (1GA8), a GalT of *Neisseria meningitidis*, for which the residues involved in the sugar–nucleotide binding and in the catalysis were well characterized. LgtC is a retaining GT that transfers α-D-Gal from UDP-Gal to a terminal lactose. The structure of LgtC was solved in complex with Mn^2+^ and a non-cleavable analog of the donor sugar (UDP-Gal) in which the hydroxyl at the 2′ position of the Gal was substituted by a fluorine for stability purposes [23]. The alignment of the protein sequences of LgtC and A111/114R-D1 (Figure 3), along with the structural superimposition of the 3D model of A111/114R-D1 based on LgtC (Appendix A) with 1GA8 (Chain A) in complex with Mn^2+^ and an analog of the donor sugar (Figure 4), clearly revealed that all the residues responsible for binding and catalysis were preserved. In detail, A111/114R-D1 shares four sugar binding residues with LgtC, namely Asp-85, Asn-149, Asp-193, and Gln-194, which correspond to Asp-103, Asn-153, Asp-188, and Gln-189 in LgtC (1GA8). Mutagenesis experiments have established that LgtC Gln-189, contained within the invariant D/EQD motif found in all members of GTs in family 8 (GT8), plays a crucial role in binding and probably in catalysis as well [24]. A catalytic mechanism leading to the retention of the anomeric carbon configuration for A111/114R-D1 is consistent with the presence in the oligosaccharide core of an α-D-Gal (Figure 1). It has also been proven that LgtC is a cation-dependent GT [23]. Indeed, LgtC exhibits the typical DXD motif (^103^DXD^105^), common in a wide range of GTs, both in prokaryotes and eukaryotes [25]. In addition, a crucial role has been attributed to Asp-103, as LgtC D103E and D103N mutants present a dramatic reduction in their activity compared to the wild-type [23].

However, the ^103^DXD^105^ motif of LgtC is not preserved in position with A111/114R-D1, except for LgtC Asp-103, which corresponds to A111/114R-D1 Asp-85. Homologous to residue LgtC Asp-103, Asp-85 of A111/114-D1 is positioned in close proximity to a divalent cation and to the nucleotide sugar, as denoted with the characteristic ligand distance < 4 A° (Table 1). It is likely that Asp-85 is sufficient in providing one side-chain oxygen ligand in coordination with Mn^2+^, as evident from the structural superimposition (Figure 4). The evidence that A111/114R-D1 possesses all residues involved in binding and catalysis is further supported by the fact that other well-noted GalTs exhibit similar homologies. The GT GlyE from *Streptococcus pneumoniae* TIGR4 (PDB: 5GVV) binds UDP-Gal [26], and it shares the same conserved sugar binding residues as LgtC and A111/114R-D1 (Asp-103, Asn-142, Asp-177, and Gln-178) (Appendix A). Mutation of these key residues in GlyE completely abolished the hydrolytic activity [26]. Additionally, the bifunctional domain polymerase WbbM from *Klebsiella pneumoniae* (PDB: 6U4B) possesses a C-terminal galactopyranosyltransferase [27] with similar homologous residues, Asp-486 and Gln-487, resembling the known GT8 family enzyme signature likely to bind UDP-Gal.

### 3.2. In Silico Analysis of A111/114R Domain 2

Three-dimensional renderings of the central domain of A111/114R (261 to 559 aa), referred to as A111/114R-D2, were assembled based on protein alignment and secondary structure similarities. Phyre2 analysis assigned protein model predictions based on as high as 97% coverage with 100% confidence (11–18% sequence identity) from multiple N-acetylgalactosaminyltransferases (GalNAcT) (Appendix A). The top ranked model was based on the second domain of the chondroitin polymerase K4CP from *E. coli* (PDB: 2Z86), namely 197 residues (66% of the protein sequence) of A111/114R-D2 were modelled with 100% confidence. K4CP is a bi-functional enzyme organized into two GT-A domains (A1 and A2) that catalyzes elongation of the bacterial chondroitin chain [28]. K4CP A1 (1–417) and A2 (418–682), located respectively to the N- and C-terminal, are engaged in the transfer of N-acetylgalactosamine (GalNAc) and glucuronic acid (GlcA) residues alternatively from UDP-GalNAc and UDP-GlcA. The 3D model of A111/114R-D2 (261–474) is based on the second domain (A2) of K4CP (435 to 631 aa), which binds UDP-GlcA, thus excluding GalNAcT activity. The GlcA, absent in the oligosaccharidic core, has the same stereochemistry of Xyl and differs from this monosaccharide by a carboxyl function attached to carbon 5. This finding suggests that the A111/114R-D2 could be a XylT, and for that reason we used K4CP-A2 as a reference to assess the conservation of the residues implicated in binding and in catalysis. Sequence alignment of A111/114R-D2 with K4CP-A2 validated the conserved residues involved with GlcA binding and divalent cation coordination (Figure 3). In detail, A111/114R-D2 residues involved in the sugar-nucleotide binding are Pro-265, Asp-294, and Asn-323, which correspond to Pro-439, Asp-469, and Asn-496 in K4CP-A2 [28]. Superposition of K4CP Chain D with the predicted A111/114R-D2 structure (Figure 4) showcase these residues aligning in close proximity (<4 Å) to the nucleotide sugar (Table 1), strengthening their participation in sugar-binding and catalysis. K4CP, in analogy with other GT-A fold GTs, has a ^519^DSD^521^ motif coordinating the Mn^2+^. This DXD motif is preserved in A111/114R-D2 and corresponds to ^351^DDD^353^.

Together, the corresponding catalytic sites and DXD motif support orthology in enzyme activity. Of further note, two additional DXD motifs (Appendix A) are present downstream of A111/114R-D2 (^422^DRD^424^ and ^439^DPD^441^) that potentially could play an active role in substrate recognition or catalysis, but they do not exhibit homology with K4CP (Figure 3). Should A111/114R-D2 have functional XylT activity, it would be the first XylT sandwiched between two domains specific for different nucleotide sugars. The placement of the proximal Xyl in the core glycan structure, positioned closely to Gal and Fuc, is agreeable with the proposed domain organization of A111/114R.

### 3.3. In Silico Analysis of A111/114R Domain 3

The C-terminal domain (560 to 860 aa), referred to as A111/114R-D3, was clearly predicted as a putative α-1,3-FucT (Appendix A) modelled from *Helicobacter pylori* (PDB: 2NZW) with 100% confidence and 15% sequence identity (86% coverage). The prediction is that Fuc is linked to the O-3 of a Glc, which, like the monosaccharide contributions of domains D1 and D2, is a component of the overall virus core glycan structure. FucT belongs to the GT B family, in which the protein contains N- and C-terminal domains binding to the acceptor and donor substrates, respectively. Normally, these GTs do not have a recognizable DXD motif responsible for the Mn^2+^/Mg^2+^ binding; however, A111/114R-D3 has a ^642^DLD^644^ signature (Appendix A) that must be evaluated for activity.

A sequence alignment of A111/114R-D3 and *H. pylori* FucT (hpFucT) revealed conserved residues involved in binding of the donor substrate, GDP-Fuc (Figure 3); Asn-240, Tyr-246, Glu-249, and Lys-250 correspond to Asn-749, Tyr-755, Glu-758, and Lys-759 in A111/114R-D3, respectively (Table 1). Classified as an inverting GT, the proposed catalytic mechanism of hpFucT incorporates Glu-95 as a general base in catalysis, while Lys-250 and Arg-195 in hpFucT share a key role in the neutralization of the negative charged phosphate groups from GDP-Fuc to facilitate the glycosidic bond cleavage [29]. Glu-249 acts to stabilize in part the positive charge developed in the transition state as well as to form two hydrogen bonds with both the ribose and the Fuc residues of GDP-Fuc. The Glu-95 residue of hpFucT is located in the N-terminal domain, presumably associated with the acceptor substrate, and it has no equivalent in A111/114R-D3. This finding could be related to alternative acceptors in the two FucT enzymes compared here, for example, hpFucT binding GlcNAc or A111/114R-D3 binding Glc.

Comparison of superimposed proteins disclosed that the second half of A111/114R-D3 traces the C-terminal domain of hpFucT Chain A (160 to 320 aa), illustrating similarities in secondary structures. In contrast, structure resemblance decreases towards the N-terminal regions, which could be a result of different acceptor substrates. The open structure of the N-terminal region of the A111/114R-D3 model could be a consequence of a larger acceptor.

To evaluate the sugar-binding and catalytic sites of A111/114R-D3, we superimposed the domain with hpFucT (Chain A) in complex with GDP-Fuc (PDB: 2NZY). This overlap confirmed that all the expected residues involved in the GDP-Fuc binding were preserved and in good orientation (Figure 4), in agreement with the sequence alignment data (Figure 3). A111/114R-D3 residues Asn-749, Tyr-755, Glu-758, Lys-759 appeared <4 Å from the nucleotide sugar in agreement with their equivalent residues in hpFucT (Figure 4). This observation suggests these A111/114R-D3 residues likely participate in GDP-Fuc binding and are critical for A111/114R-D3 activity. Specifically, Asn-749, Tyr-755, and Glu-758 likely stabilize the positive charge on the Fuc moiety like their equivalent residue in hpFucT. Positioned close in proximity to GDP-Fuc is A111/114R-D3 residue Arg-693 that corresponds to Arg-195 in hpFucT, an essential residue for enzyme activity. In fact, Ala mutants of Arg-195 or Lys-250 resulted in no detectable activity, supporting the idea that the two residues provide positive charges to interact with negatively charged GDP-Fuc [29]. The important role of Arg-693 is supported by its conservation amongst different chloroviruses.

### 3.4. In Vitro Evidence of A111/114R Hydrolytic Activity

In order to test A111/114R for GT activity, we used the bioluminescent UDP/GDP-Glo assays to detect free UDP/GDP released by GT-mediated hydrolysis of the nucleotide sugars. This allowed us to screen for nucleotide donor specificities without their target substrates (acceptor). A time course experiment with the full-length recombinant A111/114R protein in the presence of the core monosaccharides UDP-Gal, UDP-Xyl, GDP-Fuc, and Glc displayed evidence of UDP-hydrolysis (Figure 5). Glc was supplemented to simulate the Glc-Asn acceptor located on the nascent glycoprotein. The release of the UDP increased steadily over time, suggestive of GT activity by A111/114R-mediated hydrolysis.

We recombinantly expressed the full-length A111/114R protein (1 to 860 aa) and three variants with omitted regions (Figure 6a,b). A111/114R (1–397 aa) contains the complete domain 1 and approximately the first half of domain 2, A111/114R (266–860 aa) contains the complete domains 2 and 3, and A111/114R (391–860 aa) contains the second half of domain 2 and the complete domain 3. Constructs containing only individual domains were originally designed; however, their proteins did not exhibit activity. This may reflect inaccurate residue boundaries or an incorrect folding of the single domains in the absence of adjacent ones. Representative data from each assay are shown in Figure 6c,d and are represented as a ratio of the UDP or GDP measured from reactions containing the indicated GTs relative to the full-length protein-catalyzed hydrolysis, and negative controls in which no enzyme was added.

Analysis of UDP/GDP-hydrolysis by the A111/114R constructs reported in Figure 6b were especially revealing in regard to A111/114R domain assignments. Indeed, starting with the A111/114R (1–397) construct, the UDP molecules were detected only when UDP-Gal was used (Figure 6c). Given A111/114R-D2 is half omitted in A111/114R (1–397), hydrolysis of UDP-Gal implies A111/114R-D1 is a GalT. In agreement, no UDP was produced from UDP-Gal containing reactions involving A111/114R constructs devoid of the first domain, A111/114R (266–860) or A111/114R (391–860). This finding is in agreement with the bioinformatic data.

Hydrolysis of UDP-Xyl by A111/114R (266–860) suggests either A111/114R-D2 or A111/114R-D3 has XylT activity. However, UDP-Xyl was not detected in reactions involving A111/114R (1–397) or A111/114R (391–860), both of which are devoid of a complete D2 (Figure 6b). This suggests that A111/114R-D2 (260–560) harbors the XylT activity. GDP-Fuc hydrolysis was detected in reactions involving A111/114R (266–860) and A111/114R (391–860) exclusively (Figure 6d). These results suggest A111/114R-D3 is a FucT. Notably, GDP-Fuc reactions supplemented with UDP-Gal and UDP-Xyl showed elevated levels of liberated GDP, especially in the presence of A111/114R (266–860). This could be a result of improved protein folding allowed by the extension of residues and co-presence of the nucleotide sugars.

Finally, to evaluate the residues of A111/114R involved in hydrolytic activity, we constructed Ala mutants by site-directed mutagenesis (SDM) (GenScript) to target amino acids from each domain predicted to be involved in nucleotide–sugar or metal–ion binding. Three mutants were expressed, each containing two Ala substitutions inside separate domains. A111/114R-D1, -D2, and -D3 SDM constructs contained Ala mutants of Asp-85 and Gln-194, Asp-351 and Asp-353, and Arg-693 and Lys-759, respectively. In the presence of UDP-Xyl, UDP-Gal, GDP-Fuc, and Glc, the SDM of D1, D2, and D3 resulted in a significant reduction in UDP-sugar hydrolysis, lowering the activity by 95%, 90%, and 80%, respectively (Appendix A). Likewise, in the presence of the same nucleotide–sugars, GDP-Fuc hydrolysis was reduced by 90%, 70%, and 96%, respectively. This dramatic reduction in detectable nucleotide–sugar hydrolysis supports the idea that these residues are critical for enzyme activity, and that A111/114R functions best when all three domains are active.

## 4. Conclusions

The giant chloroviruses continue to challenge our understanding of canonical metabolic pathways in host–virus interplay. The identification of the atypical N-glycan structure attached to the PBCV-1 Vp54 has led to the characterization of virus-encoded enzymes involved in glycosylation independent of host-derived ER and Golgi GTs. The N-glycan’s core pentasaccharide conserved among the chloroviruses is especially noteworthy and prompted us to investigate candidate virus-encoded GTs involved in the assemblage of part or all the conserved core oligosaccharide. Results in this study establish that PBCV-1 encoded protein A111/114R has three GT domains of approximately 300 aa each. Evidence from a combination of amino acid alignments, three dimensional renderings, and GT assays indicate that the N-terminal (1 to 260 aa), central (261 to 559 aa), and the C-terminal (560 to 860 aa) regions resemble a GalT, a XylT, and a FucT, respectively. The three-dimensional protein models built by Phyre2 are predictions and were used only for the purpose of identifying potential individual domains. As with all methods for protein modeling prediction, caution should be exercised when evaluating structural elements of new enzymes that lack homology to currently deposited structures in the Protein Data Bank archives. In fact, since the percent identity between the various domains and the structures on which they were modeled was low, it was not possible to model the various domains accurately. Eventually individual protein domains must be solved by biochemical and crystallographic methods to fully reveal their catalytic mechanism.

Preliminary evidence suggests that A111/114R-D2 has XylT activity and, if biochemically confirmed, presents a new structural class of XylTs based on its limited resemblance to known XylTs. Moreover, A111/114R would be the second three-domain protein encoded by PBCV-1 that is involved in glycan synthesis. Indeed, recent studies showed that PBCV-1 protein A064R (638 aa) has three functional domains; the first two are GTs (β-L-rhamnosyltransferase and α-L-rhamnosyltransferase, respectively) and the third is a methyltransferase that methylates O-2 of the terminal α-L-Rha residue [30].

It is known that many of the chlorovirus genes encode enzymes involved in various aspects of carbohydrate metabolism. However, it remains unclear how the virus-encoded proteins are involved in the synthesis and/or assembly of the Vp54 glycan. For example, are the sugars added to Vp54 sequentially or are they synthesized independently of Vp54, possibly on a lipid carrier, and then attached to the protein en bloc? A slight variation of these two possibilities is to synthesize a core glycan(s) independently of the protein and attach it to Vp54. Additional experiments will be required to address this issue.

Importantly, the results described herein provide support that the synthesis of the PBCV-1 core glycan structure, or at least part of it, is accomplished with a multidomain enzyme encoded by the virus itself. This finding is in line with the finding that another GT of PBCV-1, the protein A064R, is able to elongate the viral glycan with two units of Rha and to methylate the ultimate unit at O-2. Taking these finding together, the dogma that all viruses use host enzymes to glycosylate their proteins is further subverted.

## Figures and Tables

**Figure 1 viruses-13-00087-f001:**
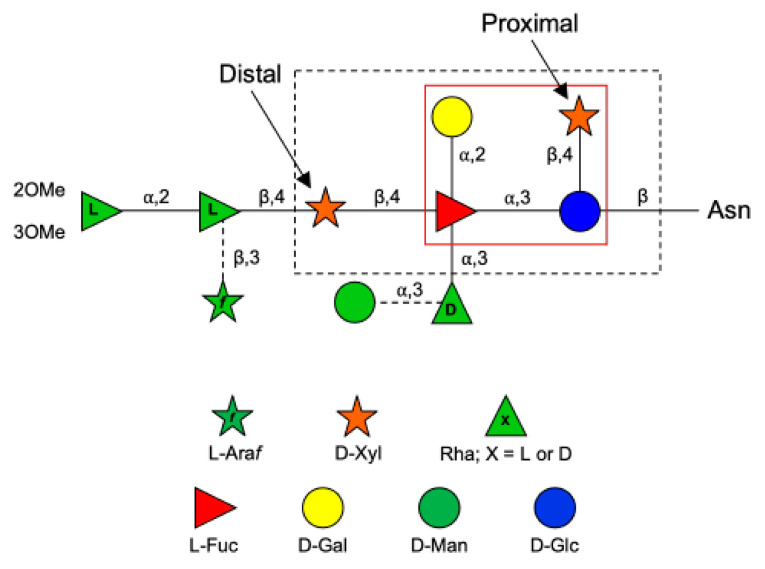
Structure of the N-glycan attached to the chlorovirus PBCV-1 major capsid protein (Vp54). Monosaccharides (Man and Ara*f*) connected by dashed lines are non-stoichiometric substituents. The larger black box encloses the conserved pentasaccharide core structure common to all chloroviruses analyzed to date. The inner red box indicates the tetrasaccharide structure of the N-glycan of the antigenic mutant chlorovirus PIL6 [6].

**Figure 2 viruses-13-00087-f002:**
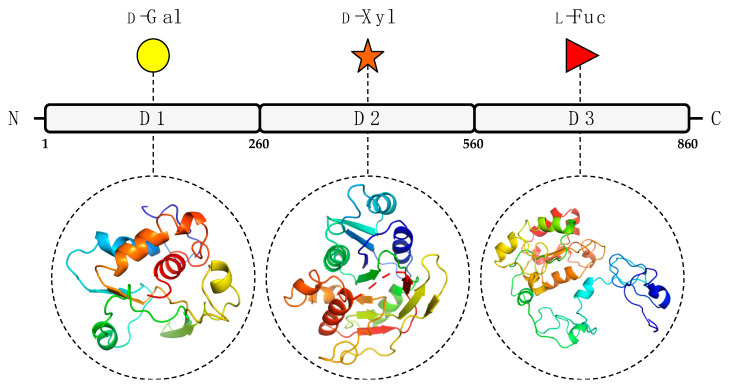
Predicted GT domains of PBCV-1 encoded protein A111/114R. A111/114R domain analysis based on remote homology identified three putative GT domains labeled as D1, D2, and D3, located at the N-terminal (1 to 260 aa), central (261 to 559 aa), and C-terminal (560 to 860 aa) regions, respectively. Below individual domains are the corresponding three-dimensional protein models assigned by Phyre2 [18] based on alignments to known protein structures identified by their PDB entry: 1GA8 Chain A (D1), 2Z86 Chain D (D2), 2NZY Chain A (D3). Protein ribbon models are rendered using rainbow colors from N-terminus (blue) to C-terminus (red). The putative domain, predicted protein model, and sugar substrate are connected by the black dashes.

**Figure 3 viruses-13-00087-f003:**
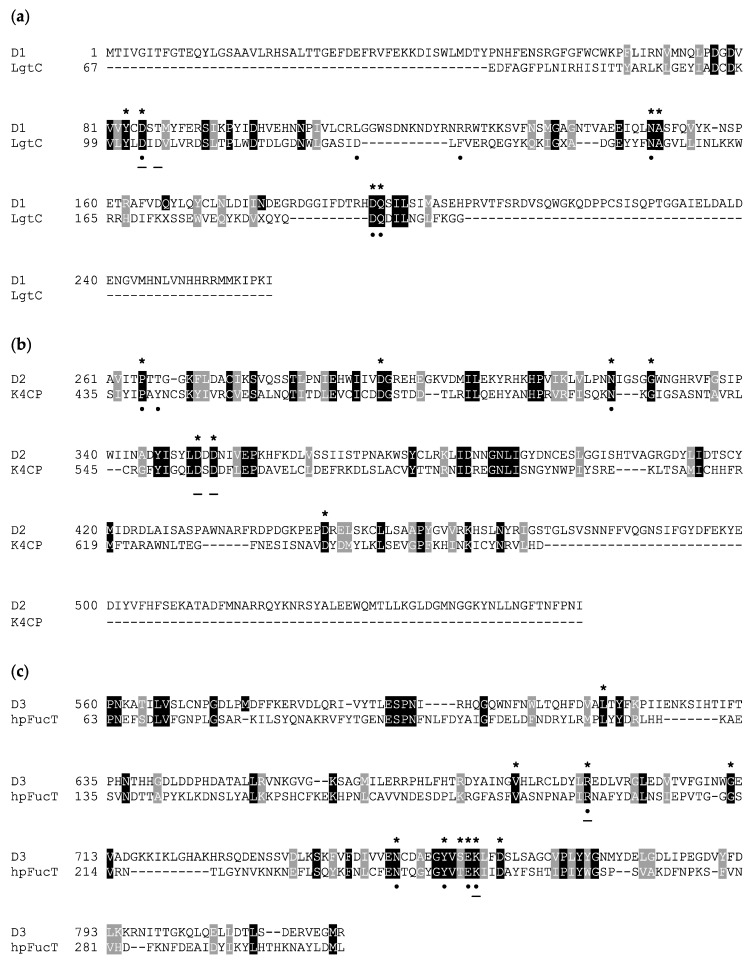
Amino acid sequence alignment of A111/114R domains and known GTs. Invariant and similar residues are highlighted in black and gray, respectively. Sequences from the following organisms were used (PDB code) for three individual domains (D1, D2, and D3), respectively: (**a**) *N. meningitidis* LgtC GalT (1GA8), (**b**) *E. coli* K4CP dual GalNAcT and GlcAT (2Z86), and (**c**) *H. pylori* FucT (2NZW). Homologous residues positioned less than 4 Å from the nucleotide sugar in the three-dimensional model analysis are marked with an asterisk (*****). Known residues from annotated enzymes that interact with the nucleotide sugar donor and ion are marked with a dot (**•**) and underline (**−**), respectively. Multiple alignment was performed by Phyre2 using structural information and homology extension. File output was compiled by BOXSHADE.

**Figure 4 viruses-13-00087-f004:**
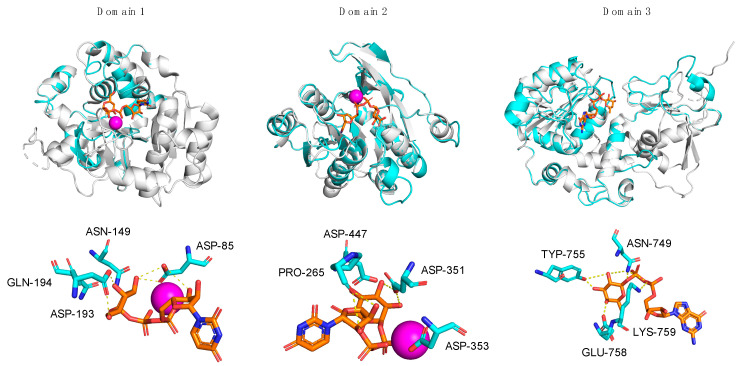
Superpositions of A111/114R-D1, -D2, and -D3 with structural homologs. Individual domains of A111/114R (cyan) are shown independently as ribbon diagrams superimposed with known GTs (gray) bound to respective nucleotide sugars drawn as stick models (orange) and Mn^2+^ ion (magenta): *N. meningitidis* LgtC with UDP-fluorogalactose (PDB: 1GA8), *E. coli* K4CP with UDP-glucuronic acid (PDB: 2Z86), and *H. pylori* FucT with GDP-Fuc (PDB: 2NZY) are superposed with D1, D2, and D3, respectively (left to right). The corresponding active sites of D1, D2, and D3 are shown magnified below the complexed stereoviews with labeled residues proposed to be involved in sugar and ion coordination. Hydrogen bonds are represented as yellow dotted lines.

**Figure 5 viruses-13-00087-f005:**
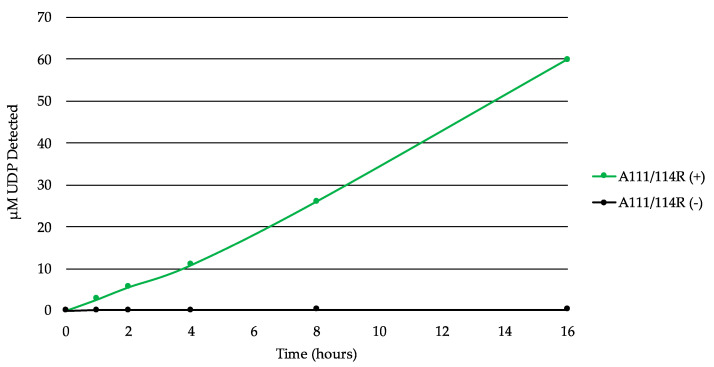
A111/114R-catalyzed hydrolysis of UDP-sugars. Time-course experiment with the full-length recombinant A111/114R protein (6 µg) in the optimized buffer consisting of 0.1 M MOPS-NaOH (pH 7.0), 10 mM MgCl_2_, and 100 µM each of UDP-Gal, UDP-Xyl, GDP-Fuc, and Glc for 16 h at 16 °C. The release of UDP was detected by the UDP-Glo^TM^ assay. Data are representative from three independent replicates, and error bars represent standard deviation.

**Figure 6 viruses-13-00087-f006:**
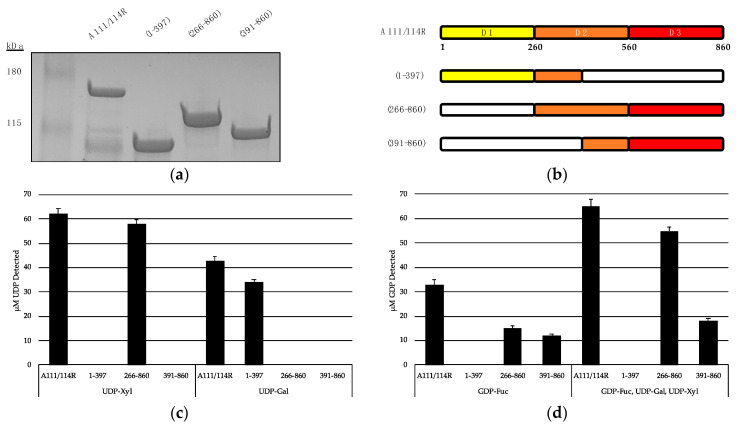
Hydrolysis of UDP- and GDP-sugars by recombinant A111/114R and truncated constructs. (**a**) SDS-PAGE analysis of the expressed proteins: full length recombinant MBP-A111/114R (144 kDa), and truncated variants MBP-A111/114R (1–397) (87 kDa), MBP-A111/114R (266–860) (110 kDa), and MBP-A111/114R (391–860) (96 kDa) were eluted from an amylose column and resolved by SDS/PAGE with Coomassie blue staining. BenchMark^TM^ pre-stained protein ladder and recombinant proteins were separated on a 4–20% tris-glycine gel. (**b**) Cartoon renderings of the full-length A111/114R protein and truncated versions are color coordinated by three putative domains (D1, D2, D3), each corresponding to a different nucleotide sugar donor. White regions denote omitted sections of A111/114R. Shortened constructs are defined by their residues in the left column. (**c** and **d**) Representative data from hydrolysis assays shown as a ratio of the UDP (**c**) or GDP (**d**) measured from reactions containing the indicated GTs relative to the negative controls where no enzyme was added. The release of UDP and GDP was detected by the UDP-Glo^TM^ assay and GDP-Glo^TM^ assay, respectively. GDP-hydrolysis from GDP-Fuc was significantly elevated in the presence of UDP-Gal and UDP-Xyl with A111/114R (266–860) and A111/114R (391–860). Data are representative from three replicates, and error bars represent standard deviation.

**Table 1 viruses-13-00087-t001:** Predicted H-bond distances between homologous A111/114R residues and nucleotide–sugar ligand.

	A111/114R Residues	H-Bond Distance of Nucleotide-Sugar (Å)
**Domain 1**	Tyr-83	<4
	Asp-85	2.6, 3.0, 3.6
	Asn-149	<4
	Ala-150	<4
	Asp-193	2.4, 2.5
	Gln-194	<4
**Domain 2**	Pro-265	2.8, 3.3
	Asp-294	<4
	Asn-323	<4
	Gly-328	<4
	Asp-351	3.2, 3.3
	Asp-353	<4
	Asp-447	<4
**Domain 3**	Leu-617	<4
	Val-684	<4
	Arg-693	<4
	Gly-711	<4
	Asn-749	1.6, 3.1, 3.3
	Tyr-755	1.9, 3.4
	Ser-757	<4
	Glu-758	2.2
	Lys-759	2.5
	Asp-762	<4

## Data Availability

Data is contained within the article or Appendix A.

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
