# Peer review of "Chlorovirus PBCV-1 Multidomain Protein A111/114R Has Three Glycosyltransferase Functions Involved in the Synthesis of Atypical N-Glycans"

_viruses, 2021, doi:10.3390/v13010087_

Round 1
Reviewer 1 Report
The manuscript by Noel et al describes the combination of genomic and bio-informatic analyses with biochemical data to ascertain that the multidomain protein A111/114R from chlorovirus PBCV-1 bears three distinct glycosyltransferase activities (GalT, XylT and FucT), responsible for the assembly of the core region of the N-glycans decorating the major capsid protein (MCP) of chloroviruses. Given that chlorovirus MCP N-glycans are very peculiar and unique, this work, trying to decipher the fundamental bases of glycan assembly is of high interest. However, experimental evidence is rather weak and the main core of the manuscript focuses on structural modelling, with a long discussion based on weak results.
Is it surprising or novel that a homology model based on a pdb entry superposes well with the given pdb entry? Phyre2 results, as every modelling, should have been taken with caution. Only about 60% of the A111/114R-D1 sequence aligns with LgtC, leading to an aborted GT-A fold (see Fig. 4a). Have the authors tried to compare the results provided by Phyre2 with those of other alignment tools (see Fig3a)? Data presented in Table 1 are obsolete. Even the astonishing CASP14 results obtained by Alphafold2 would not permit to forward hydrogen-bonding distances with this precision.
It is quite worrying to find out that constructs of individual domains did not exhibit activity – the authors could have developed on this aspect. Does A111/114R remain intact or is it post-translationally processed, ex. cleaved into individual domains. This is an important aspect with regard to access to active sites from a structural point of view. Even if experimental data are not available, the authors could have discussed this point.
Results shown in Figure 6d are interesting and might deserve some development (for the non-expert like myself). How precise are the assays used in distinguishing between UDP and GDP release? Could GDP release for construct 266-860 reveal a contribution of UDP released by D2?
Overall, the manuscript is quit fluent, but suffers at certain instances from sloppiness, with style and wording devoid of precision and leading to confusion. Starting from the title: a gene encodes proteins, but a protein does not encode proteins. I strongly invite PIs to assist revision of the manuscript.
Minor comments:
Organism names should be written in italics: Paramecium bursaria (L55), Arabidopsis thaliana (L185), Neisseria meningitidis (L199),
Abstract: “Importantly, these findings provide additional support that the chloroviruses do not use the canonical endoplasmic reticulum-Golgi glycosylation pathway “ – the authors are invited to reformulate and clearly state that they refer to the host’s Golgi-apparatus in order to avoid misunderstanding, miseducation.
Line 61 and 72: chose between non-stoichiometric and nonstoichiometric
Line 65: beta-glucose linkages per se are the most abundant glycosidic linkages in Nature (cellulose) – may be reformulate; Nature with capital N
Line 75: correct antigenetic for antigenic
Line 185: Arabidopsis thaliana
Line208: Glcn should be corrected for Gln
Line 211 and following: the statement that Gln-194, is well positioned to use its side chain nitrogen as a nucleophile is a harsh speculation. Amides are notoriously poor nucleophiles and Persson et al studying LgtC came to the conclusion that Gln 189 is unlikely to have a role as the catalytic nucleophile.
Line279 and following: this section is sloppy. The text does by no means correspond to the sequence alignment shown in Fig 3b and is mostly confusing.
Line292: is domain organization necessarily related to oligosaccharide organization? What about the distal xylose following the forwarded reasoning?
Line 298: correct GlcNAc for Glc
Line 332: Figure S3 appears to be mostly useless, given that sequence numbering in the alignment does not correspond to the numbering evoked in the text. Impossible for the general reader to spot the Arg-XX conservation.
Author Response
Dear Editor,
On behalf of all coauthors, I wish to thank the reviewers for their valuable comments that are hereafter addressed point-by-point. I hope the manuscript is now suitable for publication in Viruses.
Kind regards,
Eric Noel
Reviewer #1
Is it surprising or novel that a homology model based on a pdb entry superposes well with the given pdb entry? Phyre2 results, as every modelling, should have been taken with caution. Only about 60% of the A111/114R-D1 sequence aligns with LgtC, leading to an aborted GT-A fold (see Fig. 4a). Have the authors tried to compare the results provided by Phyre2 with those of other alignment tools (see Fig3a)? Data presented in Table 1 are obsolete. Even the astonishing CASP14 results obtained by Alphafold2 would not permit to forward hydrogen-bonding distances with this precision.
Answer: we agree with the reviewer that the presented protein models are strictly predictions and therefore should be used with caution. We added (Lines 457-459) to the manuscript to help convey this. We do not find it surprising that a three-dimensional model based on a pdb entry exhibits superposed elements with each other, however we believe the recognized homology provides reasonable guidance when predicting enzyme function. We did cross reference all Phyre2 results with multiple alignment tools and they were all in agreement. We understand and respect the reviewer’s skepticism when assigning residue proximity associated with the proposed nucleotide sugar (Table 1) based on predictive analysis. However, merit to these predictions was supported by making ala-mutant substitutions to the named residues that resulted in significant reduction in activity. This new experiment was performed after submission of our manuscript. We believe these new results would contribute to the manuscript therefore we incorporated it here (Lines 434-444 and Figure S3). In light of these new findings, we think the manuscript would benefit by keeping Table 1.
It is quite worrying to find out that constructs of individual domains did not exhibit activity – the authors could have developed on this aspect. Does A111/114R remain intact or is it post-translationally processed, ex. cleaved into individual domains. This is an important aspect with regard to access to active sites from a structural point of view. Even if experimental data are not available, the authors could have discussed this point.
Answer: It is possible A111/114R is post-translationally processed, however we believe the absence of activity from individual domains is more likely explained by inaccurate residue boundaries or an incorrect folding of the single domains in the absence of adjacent ones explained in Lines 389-390. Original domain boundaries were defined by HHpred which, like Phyre2, are predictions based on known domains. Furthermore, as reported in Figure S3, it is clear that the activity of any one domain is better if all three domains are functional, which supports the idea that the protein is not processed into three separate peptides.
Results shown in Figure 6d are interesting and might deserve some development (for the non-expert like myself). How precise are the assays used in distinguishing between UDP and GDP release? Could GDP release for construct 266-860 reveal a contribution of UDP released by D2?
Answer: UDP- and GDP-Glo assays are very sensitive to detecting hydrolysis of specific nucleotide-sugars. Negative controls were performed to verify these claims made by the assay manufacturer. And therefore, it’s unlikely enzyme-catalyzed UDP hydrolysis would be detected in a GDP-Glo assay and vice versa.
Overall, the manuscript is quit fluent, but suffers at certain instances from sloppiness, with style and wording devoid of precision and leading to confusion. Starting from the title: a gene encodes proteins, but a protein does not encode proteins. I strongly invite PIs to assist revision of the manuscript.
Answer: the title was changed as suggested.
Minor comments:
Organism names should be written in italics: Paramecium bursaria (L55), Arabidopsis thaliana (L185), Neisseria meningitidis (L199),
Answer: Paramecium bursaria (Line 57) is referenced here to define the virus name, not the ciliate alone. According to the International Committee on Taxonomy of Viruses (ITCV), virus names should not be in italics even in this case. Therefore, this was not changed. However, Arabidopsis (L199), Neisseria meningitidis (L211), and others (see Track Changes) were corrected in italics.
Abstract: “Importantly, these findings provide additional support that the chloroviruses do not use the canonical endoplasmic reticulum-Golgi glycosylation pathway “ – the authors are invited to reformulate and clearly state that they refer to the host’s Golgi-apparatus in order to avoid misunderstanding, miseducation.
Answer: this was clarified as suggested.
Line 61 and 72: chose between non-stoichiometric and nonstoichiometric
Answer: this was changed to non-stoichiometric (Line 74).
Line 65: beta-glucose linkages per se are the most abundant glycosidic linkages in Nature (cellulose) – may be reformulate; Nature with capital N
Answer: We agree with the reviewer that the beta-glucose linkage is an unabundant linkage in nature, however what is rare, is this same linkage between a glycan and a protein which is stated. Therefore, the text was left as is. Nature has been capitalized.
Line 75: correct antigenetic for antigenic
Answer: this was changed.
Line 185: Arabidopsis thaliana
Answer: this was italicized.
Line208: Glcn should be corrected for Gln
Answer: this was changed.
Line 211 and following: the statement that Gln-194, is well positioned to use its side chain nitrogen as a nucleophile is a harsh speculation. Amides are notoriously poor nucleophiles and Persson et al studying LgtC came to the conclusion that Gln 189 is unlikely to have a role as the catalytic nucleophile.
Answer: we agree with the reviewer and omitted this sentence.
Line279 and following: this section is sloppy. The text does by no means correspond to the sequence alignment shown in Fig 3b and is mostly confusing.
Answer: we agree with the reviewer and omitted this part.
Line292: is domain organization necessarily related to oligosaccharide organization? What about the distal xylose following the forwarded reasoning?
Answer: We believe so, however there is no evidence of this yet. For the other monosaccharides, like the distal xylose, these are recruited by other virus-encoded GTs.
Line 298: correct GlcNAc for Glc
Answer: this was changed.
Line 332: Figure S3 appears to be mostly useless, given that sequence numbering in the alignment does not correspond to the numbering evoked in the text. Impossible for the general reader to spot the Arg-XX conservation.
Answer: We agree with the reviewer that the original Figure S3 is unnecessary and therefore was omitted. However, we substituted the original figure with a new Figure S3. It is now accurately referenced in the supplemental section (Lines 488 and 489).
Other Notes:
We added an acknowledgement (Line 499).
References cited – Several manual edits were made due to former errors in autoformatting by EndNote. For example, authors referenced as “et al.” were corrected to include the full name of all contributing authors. Also, font formatting and journal abbreviations were made uniform.
A new Figure S3 was added.
Reviewer 2 Report
In this manuscript, Noel et al. used protein modeling and in vitro assay to determine the PBCV-1 encoded A111/114R protein has potential three glycosyltransferase domains resemble a GalT, a XylT, and a FucT domain. The results in this study support that Chloroviruses encode their own glycosyltransferase system independent of the host ER-Golgi secretory pathway for protein glycosylation. I have a major concern listed below.
Authors should include the full-length A111/114R data by UDP-Glo and GDP-Glo assays in the bar graph of Fig.6(C) and 6(D).
Author Response
Dear Editor,
On behalf of all coauthors, I wish to thank the reviewers for their valuable comments that are hereafter addressed point-by-point. I hope the manuscript is now suitable for publication in Viruses.
Kind regards,
Eric Noel
Reviewer #2
In this manuscript, Noel et al. used protein modeling and in vitro assay to determine the PBCV-1 encoded A111/114R protein has potential three glycosyltransferase domains resemble a GalT, a XylT, and a FucT domain. The results in this study support that Chloroviruses encode their own glycosyltransferase system independent of the host ER-Golgi secretory pathway for protein glycosylation. I have a major concern listed below.
Authors should include the full-length A111/114R data by UDP-Glo and GDP-Glo assays in the bar graph of Fig.6(C) and 6(D).
Answer: We agree with the reviewer. We have included the full-length A111/114R data in Figure 6C and 6D.
Other Notes:
We added an acknowledgement (Line 499).
References cited – Several manual edits were made due to former errors in autoformatting by EndNote. For example, authors referenced as “et al.” were corrected to include the full name of all contributing authors. Also, font formatting and journal abbreviations were made uniform.
A new Figure S3 was added.
Round 2
Reviewer 1 Report
The authors addressed all questions and points raised by this reviewer. They added supplementary experimental evidence in the form of site-directed mutagenesis which helps to consolidate the study presented in this manuscript.
The new Figure S3 is not accessible on the online submission page, but I have faith that the authors will provide all relevant files for final submission.
Reviewer 2 Report
The manuscript looks fine after revision.